# Proliferation and Morphological Assessment of Human Periodontal Ligament Fibroblast towards Bovine Pericardium Membranes: An In Vitro Study

**DOI:** 10.3390/ma15238284

**Published:** 2022-11-22

**Authors:** Serena Bianchi, Sara Bernardi, Davide Simeone, Diana Torge, Guido Macchiarelli, Enrico Marchetti

**Affiliations:** Department of Life, Health and Environmental Sciences, University of L’Aquila, 67100 L’Aquila, Italy

**Keywords:** cell adhesion, collagen membrane, PDL fibroblast

## Abstract

Over the past decade regenerative branches of dentistry have taken on more and more importance, resulting in the development of performing scaffold materials. These should induce cell adhesion, support, and guide the tissues’ growth. Among the developed materials, we can include resorbable or non-membranes. The purpose of this study was to investigate the proliferation abilities and the attachment of human periodontal ligament fibroblasts (HPLIFs) over two bovine pericardium membranes with different thicknesses, 0.2 mm and 0.4 mm, respectively. These membranes have been decellularized by the manufacturer, preserving the three-dimensional collagen’s structure. The HPLFs were cultured in standard conditions and exposed to the tested materials. XTT was performed to assess cell proliferation, while light microscopy (LM) and scanning electron microscopy (SEM) observations assessed fibroblast morphology at different times (T1, T2, and T3). Proliferation assays have shown a statistically significant difference in growth at T1 (*p* < 0.05) in the cells cultured with a thicker membrane compared to the thinner one. LM analysis showed healthy fibroblasts in contact with the membranes, appearing larger and with a polygonal shape. SEM observation demonstrated thickening of the fibroblasts which continued to adhere to the membrane’s surface, with enlarged polygonal shape and developed filipodia and lamellipodia. These results showed a similar cell behavior over the two bovine pericardium membranes, demonstrating a cellular migration along and within the layers of the membrane, binding with membrane fibers by means of filopodial extensions. Knowledge of the effects of the collagen membranes derived from bovine pericardium on cellular behavior will help clinicians choose the type of scaffolds according to the required clinical situation.

## 1. Introduction

Regenerative branches of dentistry have stimulated the development of conductive and inductive biomaterials [1,2] used in various techniques such as sinus augmentation, ridge preservation, guided bone regeneration (GBR) and periodontally regenerative procedures.

Among these, the GBR, which has the aim of improving and guiding the mechanism of bone repair, requires scaffold materials with a three-dimensional structure, similar to the extracellular matrix. Biomaterials are a class of materials, characterized by unique chemical, mechanical and biological properties, mainly osteoinductivity and osteoconductivity, which make them suitable and safe to interact with living tissue [3,4]. The first application of biomaterials was during the 1950s, but only in the 1970s was there an onset of the second generation of materials, for human tissue replacement, and their evolution induced a significant impact on regenerative medicine [3].

The main hallmark of scaffold materials is to induce cell adhesion, enhance specific protein synthesis, and support the growth of bone tissue. In GBR procedures, there is also the need to exclude epithelial and connective tissue cells from the wound area to be regenerated, so that only bone cells can grow in the defective area.

Therefore, non-resorbable or bioabsorbable membranes are essential to properly guide the regeneration process [5].

Four common non-resorbable membranes are available in GBR procedures: expanded-polytetrafluoroethylene (e-PTFE); dense-polytetrafluoroethylene (d-PTFE); titanium mesh; and titanium-reinforced PTFE [6,7,8,9].

The e-PTFE membrane presents many small pores, facilitating cellular attachment, preserving the wound area, and inhibiting the migration of connective and epithelial cells [8]. As with all non-resorbable membranes, the drawback of the use of e-PTFE is the high risk of infection in case of exposure to the oral cavity.

The d-PTFE membranes present smaller pore sizes than the e-PTFE, reducing the possibility of microbial colonization. Beyond allowing the maintenance of the space and the stabilization of the wound, the d-PTFE membrane is removable with minimal intervention [10].

Titanium mesh and titanium-reinforced PTFE membranes are rigid, which is fundamental for stability and space maintenance. In case of exposure, there is a minimal risk of infection.

In addition, non-resorbable membranes require a primary fixation in the first surgery and a second surgical procedure for the removal; moreover, their use requires an important learning curve from the operator, which increases the patient’s discomfort and the risk of secondary infection [5].

Bioresorbable membranes, made of collagen, are employed as an alternative. The collagen can derive from human, porcine, or bovine pericardium, calfskin, Achilles tendon, and dermis. Pitaru et al. demonstrated that type-I collagen membranes prevent the apical migration of epithelium in a canine model and facilitate new connective tissue attachment and regeneration. In the host environment, then, neutrophils, monocytes, and fibroblasts release matrix metalloproteinases (MMPs) to resorb the collagen matrix [11].

Bovine and porcine-derived collagen types 1 and 3 are the resorbable membranes most frequently used in clinical routine; an example is the bovine pericardium membranes, which are decellularized to remove antigenic epitopes associated with cell membranes and intracellular components, thus to enhance the biocompatibility to the wound site. According to Liu et al., the decellularization method critically affects recellularization potential [4]. What is preserved is the three-dimensional structure of the collagen net.

Bovine pericardium membranes showed significant functional and morphological potential, which provides the opportunity to examine cellular behavior, also in different fields [4]. Several studies in fact have successfully detected positive histological findings, due to the use of bovine pericardium membranes [12]. Furthermore, Athar et al. report the potential role of lyophilized bovine pericardium, in periodontal ligament fibroblast attachment, migration, and proliferation [13]. Other recent studies have reported an osteoangiogenesis differentiation process in periodontal ligament stem cells due to the interaction with bovine pericardium membranes [14]. Recently, the bovine pericardium membrane was shown to be efficient for GBR for mandibular rabbit defects [15], but before further clinical applications, in vivo studies on larger animals should be performed.

The aim of the study is to evaluate the reaction and the attractivity of human periodontal ligament fibroblasts (HPLFs) cultures toward bovine pericardium membranes of different thicknesses.

## 2. Materials and Methods

The study design included the evaluation of HPLFs proliferation in contact with bovine pericardium membranes by means of XTT essay and the morphology of HPLFs in contact with bovine pericardium membranes by means of light and scanning electron microscopy, as previously described [12,13,14].

### 2.1. Membranes

The membranes chosen for this study were derived from bovine pericardium, decellularized by the manufacturer (UBGEN Srl, Vigonza (PD), Italy); the three-dimensional structure of collagen is maintained according to the manufacturer’s specifications. The assays were performed using two membrane thicknesses: 0.2 mm and 0.4 mm, henceforth referred to as X1 and X2.

### 2.2. Cell Culture

The HPLFs (Innoprot, Bizkaia, Spain), after the thawing, were cultured according to the manufacturer’s instructions [15]. The original vial containing 5 × 10^5^ cells in 1 mL of volume was cultured in three plastic culture dishes in fibroblast medium (Innoprot, Bizkaia-Spain) and incubated under standard cell culture conditions (37 °C in 5% CO_2_). According to the manufacturer, a bottle of fibroblast medium is composed of 500 mL of basal medium, 10 mL of fetal bovine serum (FBS—Innoprot, Bizkaia-Spain), 5 mL of fibroblast growth supplement, and 5 mL of penicillin/streptomycin solution (10,000 IU/mL of penicillin, 10 μg/mL streptomycin—Innoprot, Bizkaia, Spain) [16]. Once the cells reached sub-confluence, they were detached using 0.05% trypsin and subcultured at 110 cells/mm density. The cells were used at subculture passages 7 or 8 for all experimental assays.

### 2.3. Cell Proliferation Assays and Statistical Analysis

The cell proliferation assay was performed according to the ISO EN 10993-5 standard [17]. Briefly, 2 mm x 3 mm of each sterile sample was placed into 24-well plates. According to the protocol procedure, 1 mL of fibroblast medium was added (henceforth referred to as “extraction medium”) to each well and incubated for 24 h at 37 °C. Then, under standard cell culture conditions, 10^3^ cells per well were seeded in a 100 µL of extraction medium. The negative control group was seeded in 100 µL of fibroblast medium. The XTT assay (Cayman Chemical, Ann Arbor, MI, USA) allowed observation of the fibroblasts’ starting condition (T0) and proliferation activity at 24 h (T1), 72 h (T2), and 7 days (T3) follow-ups at an absorbance wavelength of 450 nm. XTT tests were performed with three technical replicates.

The statistical analysis of the data included a first assessment of the normal distribution of the data, followed by a two-way analysis of variance (ANOVA) test and Dunnett’s and Bonferroni’s post hoc analyses for multiple comparisons, to check any significant variation through the considered follow-ups.

Statistical analysis and graphs were performed using GraphPad Prism 9.1.1 (GraphPad Software (version 9.1.1), San Diego, CA, USA).

### 2.4. Morphological Analysis

#### 2.4.1. Morphological Analysis—LM

As in previous studies [12,13,14], cells were plated in 60-mm diameter plastic culture dishes together with the X1 and X2 samples. The dishes were incubated under cell culture conditions. At T1, T2, and T3 the plates containing the tested memebranes and the negative control were randomly observed by means of a phase-contrast light microscope (ZEISS Primovert, Jena, Germany).

ZEISS Axiocam 208 color camera was used to capture the images at 10× and 20×.

#### 2.4.2. Morphological Analysis—SEM

As in previous studies [12,13,14], HPLFs were seeded in petri dishes containing cover glasses together with the X1 and X2 samples the tested materials. Petri dishes were seeded without membranes to serve as a negative control. At T1, T2, and T3 the cells were fixed using a 2% solution of glutaraldehyde and were processed for the SEM. Briefly, after the post-fixation in 1% osmium for 1 h, the samples were dehydrated in ascending concentration ethanol solutions of 70%, 80%, 90%, and three times 100% for 10 min each. Subsequently, the samples were immersed for 3 min in 100% HDMS (Sigma-Aldrich Srl, Milan, Italy) and air-dried. The samples were then transferred into a desiccator for 25 min to prevent water contamination. The samples were mounted on metal stubs, gold stained, and then observed by SEM (GEMINI_SEM, Zeiss, Oberkochen, Germany) at different magnifications using secondary electrons probes.

## 3. Results

The combination of XTT assays (assessing the proliferation) and the LM and SEM observation (assessing any cellular morphological change) allowed us to obtain a full and integrated overview of the performance of the examined membranes and the interaction of HPLFs.

### 3.1. Cell Proliferation Assays and Statistical Analysis

The obtained and analyzed data from the XTT showed a continued growth curve in the tested membranes (X1 and X2) and negative control (no material) as shown in Figure 1. The subsequent two-way ANOVA, which considered the variation in the optical density (OD) from the T0 and T1, T2, and T3 follow-ups, was statistically significant (*p* < 0.05).

Dunnett’s post hoc analysis did not show a significant variation in the growth curve between the negative control and all experimental membranes from the T0 to T3. The post hoc Bonferroni analysis showed a significant variation in the growth at T1 follow-up between X1 and X2 (Table 1).

### 3.2. Morphological Analysis—LM

In all observed cultures, the LM observations showed a uniform layer of healthy cells. However, morphological differences between them were appreciable (Figure 2).

At T1 of culture, the cell shape of the control group is fusiform morphology and of small dimension. Bothe the cells of X1 and X2 groups present a morphology larger than the negative control cells.

At T2 of culture, the cell shape of the control group continues to maintain appears fusiform. Observations showed a densely packed layer of fibroblasts. The X1 and X2 fibroblasts instead present an enlarged morphology and have lost the fusiform shape, presenting polygonal traits.

At T3 of culture, the control group fibroblasts showed a layer of fusiform, density, and packed cells. The cultures in contact with X1 and X2 membranes, beyond being packed and dense, appeared large and polygonal.

### 3.3. Morphological Analysis—SEM

All the observed groups (X1, X2, and negative control) appeared with abundant and healthy cells at the SEM analysis (Figure 3), but with morphological changes.

At the T1 follow-up, the fibroblast of the control group showed flattened and elongated, signs of health. The fibroblast therefore displayed signs of developing lamellipodia due to the presence of cytoplasmic digitations.

The body of fibroblasts exposed to membranes X1 and X2 appeared to adhere to the membrane surfaces. The cells’ shape is not fusiform but enlarged and presenting cytoplasmatic extensions and digitation on the cellular membrane.

At the T2 follow-up, the morphological features of the control group overlapped with the T1 group. Both X1 and X2 samples showed fibroblasts thickened and continuing to adhere to the membrane surfaces, with enlarged polygonal shapes, and developed filipodia and lamellipodia.

At the T3 follow-up, the fibroblasts of the control group appeared to form an extensive cellular network. In both, the X1 and X2 samples, lamellipodia and filopodia appeared more abundant. The fibroblast morphology appeared dynamic and with an intimate relationship between cell digitations and the elements of the membranes.

## 4. Discussion

The GBR and grafting techniques were used as the first device impractical millipore filter barriers [16]. EPTFE non-resorbable membranes, first used in 1984, reported successful outcomes, which were, however, compromised by microbial colonization. In addition, the necessary second surgical intervention for barrier removal represents a further limitation to this use [17]. The developed resorbable membranes, animal-derived or constituted by synthetic polymers, avoid these limitations [16]. Indeed, they are gradually hydrolyzed or enzymatically degraded [18] without the necessity of a second surgical stage. The fabrication sources varied from rat or cow collagen [17], polylactic acid [19], polyglycolide, and acellular dermal allograft [20], to freeze-dried dura mater [21].

The bovine pericardium has historically been used for regenerative and prosthetic purposes, such as the realization of valve prosthetics and repair patches [13]. Indeed, the bovine pericardium is characterized by a collagen-rich extracellular matrix, a natural and hospitable microenvironment, favoring the migration and proliferation, and therefore regeneration, of cells [22,23,24]. In addition, the collagen membranes are flexible and handy during surgery, absorbing blood clots, and if they present fibers of collagen type I, cell growth and migration are improved [24]. All these properties result in the realization of the best environmental conditions for regenerative surgery.

The physical characteristics of membranes, such as porosity, surface topography, chemical composition, stiffness, and membrane barrier, can influence guided tissue or guided bone regeneration [25]. Moreover, as a living tissue shows an innate ability to remodel, collect, adapt, and retrieve information, sometimes an artificial scaffold may not be able to mimic all these functions [26]. An essential role in the design of artificial scaffolds is reserved for the biochemical pathways, which underlie cell and biomaterial interaction, fundamental for developing a bioactive scaffold [27]. Simultaneously, material design should also contemplate the immune rejection and the inflammatory response, which play, together with growth factors and cytokine, a pivotal role in the cell interaction environment [28].

In GBR procedures, gingival tissue enhancement and adequate space maintenance are parameters fundamental for periodontal repair [29,30] Therefore, the membrane thickness should range from 0.2 to 0.5 mm [30] to provide adequate thickness [31] for tissue enhancement and space maintenance required for periodontal repair process [29]. Specifically, membranes 0.2 mm thick facilitate soft tissue manipulation and 0.4 mm is the material of choice among early-generation membranes [31].

The bovine derivation requires the decellularization of the bovine pericardium for use in the human body [32]. The ideal treatment, beyond guaranteeing the safe use of the product, should help to reach the necessary physical properties such as thickness, tensile strength, elasticity, nontoxicity, and low calcification, for a performant membrane [32].

Periodontal soft tissues such as gingiva play protective and support roles in infection and/or trauma cases. The fibroblasts of periodontal structures produce and organize collagens, fibronectin, and other proteoglycans [33], greatly contributing to the tissue repair mechanisms.

From this perspective, this study aims to answer two significant questions about the interaction between HPLFs and bovine pericardium membranes: from a cellular point of view, how these biomembranes might activate proliferative patterns [4] in this cellular type, and from a morphological point of view, how they might influence their morphology. These are the key points of this study, which might help to understand the proliferative abilities and the attachment of HPLFs to bovine pericardium membranes, according by different thicknesses. This work describes the proliferation and the morphological changes of periodontal ligament fibroblasts over two collagen type I bovine pericardium membranes, with different thicknesses. The proliferative and morphological assays showed that the fibroblasts’ morphology and proliferation were similar between experimental groups and negative control. The XTT assays found the tested membranes to be highly biocompatible and stimulating cellular proliferation. In particular, the wells containing the HPLFs exposed to the thicker membranes showed statistically significant proliferative data at the T1 follow-up more so than the well exposed to the X1 membrane. This proliferation data agrees with similar data in the literature [32].

Indeed, Ngoc Nguyen et al., in an in vitro study, assessed how the membrane derived from bovine pericardium induced and stimulated attachment, migration, and proliferation of human gingival fibroblasts [34]. The authors demonstrated how the extracellular matrix of the bovine pericardium membrane can provide adequate support, acting as a cell scaffold [31,33]. As bovine pericardium membranes showed significant biocompatibility with human fibroblasts, by stimulating their migration patterns they may be successfully applied as a guided biomembrane in periodontal reconstruction and regeneration, as suggested by several recent studies on this topic [35].

SEM observation through the different follow-ups showed that HPLF were flattened, with a polygonal-shaped appearance on the bovine pericardium membrane, a sign of cellular health [36]. In addition, the HPLF showed a bind with collagen fibers due to the presence and development of lamellipodia and filopodia.

These results agree with Berahim et al. (2011); in their work, fibroblasts were attached, flattened, and able to migrate over and into porcine collagen membranes (BioGuide) [37].

In addition, Atar et al. [13], similarly to our study, found that HPDLFs seeded on bovine pericardium membranes showed flattened and developed cellular processes at SEM observations, improving the chances of using bovine pericardium as appropriate material in periodontal regenerative procedures.

Earlier studies have described the flattened stellate shape of fibroblasts as indicative of cellular health, while lamellipodia and filopodial extensions bringing about cytoskeletal organization and migration are integral to their regenerative function [37,38].

The positive outcomes of GBR or GTR interventions rely on the proliferation and differentiation of the undifferentiated cells at the surgery site. Indeed, a rapid interaction of the cellular proteins with the biomaterials and high cellular differentiation correspond to fast proliferation [39].

To the best of our knowledge, there is little evidence in the literature about the possible role of membrane surface morphology implied in cellular genetic expression, differentiation, migration, and proliferation. Li et al. [24], studied the effects of microgrooved collagen membranes on mesenchymal stem cells. The reported observations concluded that microgrooves significantly affect the alignment, morphology, and collagen synthesis of the cells [27].

The dynamic morphology of the HPDLFs observed by LM and SEM confirmed how the surface morphological and physical properties influence cellular behavior. As it is well acknowledged, surface roughness, wettability, and energy constitute fundamental parameters that influence membrane performances.

Open-porous, rough, and chemically activated membrane surfaces favor direct protein adsorption, especially fibronectin and albumin [40], which influence adhesion and cellular proliferation [41].

## 5. Strengths and Limitations

This study investigated HPLF reaction to bovine pericardium membranes, characterized by different thicknesses. It provides a morphological characterization of this cellular type, by describing its structural features, as a significant inception of this research. The use of other microscopy techniques to understand the interaction between biomembranes and HPLFs, such as confocal microscopy (CM) analysis and transmission electron microscopy (TEM) evidence, may provide a morpho-functional overview of the interaction between these cells and applied membranes. Due to the lack of ultrastructural evidence on bovine pericardium membranes on HPLFs, it will be useful in the future to also perform an ultrastructural analysis by TEM, allowing the visualization of representative ultrastructural details.

## 6. Conclusions

The results from our electron microscope study demonstrated similar cell behavior over the bovine pericardium membrane. Furthermore, the cells showed migration along and within the layers of the membrane along with binding to membrane fibers by means of filopodial extensions.

## Figures and Tables

**Figure 1 materials-15-08284-f001:**
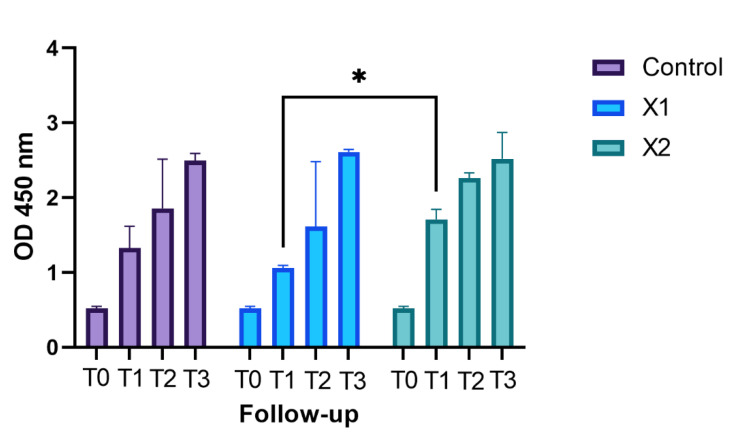
The above separated graph bar shows the grouped data of mean and SD of the OD values (Y-axis) of the cell not exposed to materials (Control) and exposed to the different membranes (X1 and X2) at the different follow-up times (T0, T1, T2, and T3). The * symbol indicates the Bonferroni multiple comparison test which assessed a difference between X1 and X2 at T1 statistically significant.

**Figure 2 materials-15-08284-f002:**
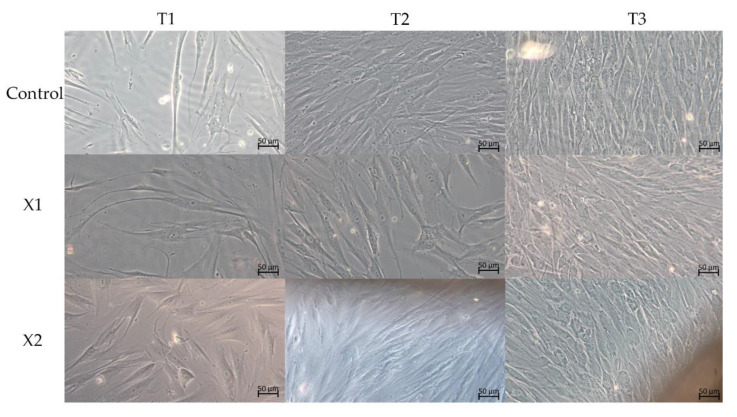
Contrast-phase light microscopy images of HPLF cells with the examined materials and controls at the different examined times. 20× magnification.

**Figure 3 materials-15-08284-f003:**
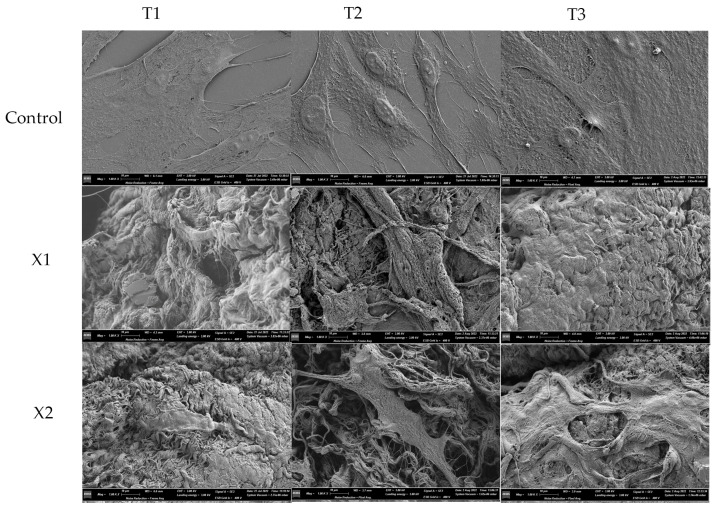
SEM images of HPLF cells with the examined materials and the controls at the different examined times. Magnification 1000×.

**Table 1 materials-15-08284-t001:** Bonferroni’s post hoc multiple comparison results. N.S. = not significant.

Bonferroni’s Multiple Comparisons Test			
T0	Mean Diff.	95.00% CI of Diff.	*p* Value
Control vs. X1	0	−0.09880 to 0.09880	N.S.
Control vs. X2	0	−0.09880 to 0.09880	N.S.
X1 vs. X2	0	−0.09880 to 0.09880	N.S.
**T1**			
Control vs. X1	0.2643	−0.9920 to 1.521	N.S.
Control vs. X2	−0.3857	−1.347 to 0.5760	N.S.
X1 vs. X2	−0.65	−1.172 to −0.1279	<0.05
**T2**			
Control vs. X1	0.248	−2.358 to 2.854	N.S.
Control vs. X2	−0.4037	−3.223 to 2.416	N.S.
X1 vs. X2	−0.6517	−4.440 to 3.136	N.S.
**T3**			
Control vs. X1	−0.1167	−0.4698 to 0.2365	N.S.
Control vs. X2	−0.025	−1.353 to 1.303	N.S.
X1 vs. X2	0.09167	−1.424 to 1.607	N.S.

## Data Availability

Data will be available from the corresponding author upon reasonable request.

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
