# Peer review of "Proliferation and Morphological Assessment of Human Periodontal Ligament Fibroblast towards Bovine Pericardium Membranes: An In Vitro Study"

_materials, 2022, doi:10.3390/ma15238284_

Round 1

Reviewer 1 Report

The paper entitled “Proliferation and Morphological assessment of human periodontal ligament fibroblast towards bovine pericardium membranes: an in vitro study” investigates the proliferation of human periodontal fibroblasts over two bovine pericardium membranes. The paper is interesting and is generally well-written. The manuscript can be accepted for publication with the following minor revisions.

1.       There is a spurious full stop at the end of the title. This is not usually a common practice.

2.       Can the authors highlight the key observations from discussing the comparing the results observed in this study?

3.       What are the challenges of developing artificial scaffolds for this application? Can the authors elaborate on the discussion in this regard?

4.       What are the limitations and prospects of this study? it is recommended that the authors add a new section in this regard before the conclusion.

5.       The introduction can be improved by reviewing key literature in the field. Some suggestions are below.

Future Directions and Requirements for Tissue Engineering Biomaterials. Encyclopedia of Smart Materials, Volume 1, 2022, Pages 195-218.

Effect of bovine pericardial extracellular matrix scaffold niche on seeded human mesenchymal stem cell function. Scientific Reports volume 6, Article number: 37089 (2016).

Author Response

Resubmission of manuscript ID:  materials - 2025959.

Thank you very much for the consideration You gave to the submitted manuscript. Here the changes list according to the reviewer’s suggestions.  They have helped us to improve the content, clarity, and impact of the manuscript. Thanking you very much for your kind attention and consideration I send you, also on behalf of the other co-authors, our best cordial regards.

Sincerely yours

Prof. Enrico Marchetti

Reviewer 1

The paper entitled “Proliferation and Morphological assessment of human periodontal ligament fibroblast towards pericardium membranes: an in vitro study” investigates the proliferation of human periodontal fibroblasts over two bovine pericardium membranes. The paper is interesting and is generally well-written. The manuscript can be accepted for publication with the following minor revisions.

We firstly would like to thank the reviewer for these valuable and constructive comments and suggestions. We addressed each of them in the revised manuscript. New and/or modified sections in the revised version of manuscript are highlighted in the text.

Here we provide a detailed point-by-point reply.

There is a spurious full stop at the end of the title. This is not usually a common practice.

We would like to thank the reviewer for this comment. The paper has been carefully revised and edited to improve the grammar, syntax, and punctuation. Specifically, the full stop has been deleted, according to reviewer’s suggestion.

Can the authors highlight the key observations from discussing the comparing the results observed in the study?

We would like to thank the reviewer for constructive comments and suggestions. We optimized the discussion, by highlighting the key points of this study, in terms of proliferation and morphological changes.

Lines 254-260

What are the challenges of the developing artificial scaffolds for this application? Can the authors elaborate on the discussion in this regard?

We thank the reviewer for this precious and constructive comment. In the discussion, we also refer to the most significant challenges in the production of artificial scaffolds and their impact on cell and tissues interaction. We also add new reference, in this section:

Tibbit et al 2015

Kim et al 2017

Urciuolo and De Coppi 2018

Lines 232-238

What are the limitations and prospects of this study? It is recommended that the authors add a new section in this regard before the conclusion.

We would like to thank the reviewer and we are grateful for this comment. We have added a section, about the strengths and limitations of our study.

This study investigates HPLF reaction to bovine pericardium membranes, characterized by different thicknesses. This study provides a morphological characterization of this cellular type, by describing its structural features, as a significant inception of this research. The use of other microscopy technique to understand the interaction between bio membranes and HPLFs, such as confocal microscopy (CM) analysis and transmission electron microscopy (TEM) evidence, may provide a morpho - functional overview of the interaction between these cells and applied membranes. Due to the lack of ultrastructural evidence on bovine pericardium membranes on HPLFs, it will be useful in the future to perform also an ultrastructural analysis by TEM, allowing the visualization of representative ultrastructural details.

The introduction can be improved by reviewing key literature in the field. Some suggestions are below.

We would like to thank the reviewer and we are grateful for this comment. We improved the introductive section, by including the suggested references(now 3 and 4).

“Biomaterials are a class of materials, characterized by unique chemical, mechanical and biological properties, mainly osteinductivity and osteoconductivity, which make them suitable and safe to interact with a living tissue [3,4]. The first application of biomaterials was during 1950s, but only in 1970s there was the onset of a second generation of materials, for human tissue replacement, and their evolution induced a significant impact on regenerative medicine[3].   “

Reviewer 2 Report

The topic is interesting and important to warrant publication.

The manuscript is written well, properly organized, and easy to follow.

The information is presented in an open-minded and objective manner.
The abstract is concise and clear.
M&M are well presented.

Conclusions are presented according to the results.
English is clear and easy to understand.

However, there are some comments and recommendation to be mentioned.

Although the purpose of this study is to investigate the proliferation abilities of HPLIFs  over two membranes with different thicknesses, 0.2mm and 0.4mm, there is no clear statement about which thickness of the membrane is better to use 0.2 mm or 0.4 mm  in the discussion part or in the conclusion part ?

In the introduction part, it is suggested to add some similar studies as a reference and to give some related background history

Also why you didn’t add some clinical studies that have used the same type of membrane with the same thickness?

I recommend suggesting some ideas for future research on the same topic or at the clinical level.
